# VAIN: Attentional Multi-agent Predictive Modeling

**Yedid Hoshen**
Facebook AI Research, NYC
yedidh@fb.com

## Abstract

Multi-agent predictive modeling is an essential step for understanding physical, social and team-play systems. Recently, Interaction Networks (INs) were proposed for the task of modeling multi-agent physical systems. One of the drawbacks of INs is scaling with the number of interactions in the system (typically quadratic or higher order in the number of agents). In this paper we introduce VAIN, a novel attentional architecture for multi-agent predictive modeling that scales linearly with the number of agents. We show that VAIN is effective for multi-agent predictive modeling. Our method is evaluated on tasks from challenging multi-agent prediction domains: chess and soccer, and outperforms competing multi-agent approaches.

## 1 Introduction

Modeling multi-agent interactions is essential for understanding the world. The physical world is governed by (relatively) well-understood multi-agent interactions including fundamental forces (e.g. gravitational attraction, electrostatic interactions) as well as more macroscopic phenomena (electrical conductors and insulators, astrophysics). The social world is also governed by multi-agent interactions (e.g. psychology and economics) which are often imperfectly understood. Games such as Chess or Go have simple and well defined rules but move dynamics are governed by very complex policies. Modeling and inference of multi-agent interaction from observational data is therefore an important step towards machine intelligence.

Deep Neural Networks (DNNs) have had much success in machine perception e.g. Computer Vision [1, 2, 3], Natural Language Processing [4] and Speech Recognition [5, 6]. These problems usually have temporal and/or spatial structure, which makes them amenable to particular neural architectures - Convolutional and Recurrent Neural Networks (CNN [7] and RNN [8]). Multi-agent interactions are different from machine perception in several ways:

- The data is no longer sampled on a spatial or temporal grid.
- The number of agents changes frequently.
- Systems are quite heterogeneous, there is not a canonical large network that can be used for finetuning.
- Multi-agent systems have an obvious factorization (into point agents), whereas signals such as images and speech do not.

To model simple interactions in a physics simulation context, Interaction Networks (INs) were proposed by Battaglia et al. [9]. Interaction networks model each interaction in the physical interaction graph (e.g. force between every two gravitating bodies) by a neural network. By the additive sum of the vector outputs of all the interactions, a global interaction vector is obtained. The global interaction alongside object features are then used to predict the future velocity of the object. It was shown that Interaction Networks can be trained for different numbers of physical agents

and generate accurate results for simple physical scenarios in which the nature of the interaction is additive and binary (i.e. pairwise interaction between two agents) and while the number of agents is small.

Although Interaction Networks are suitable for the physical domain for which they were introduced, they have significant drawbacks that prevent them from being efficiently extensible to general multi-agent interaction scenarios. The network complexity is $O(N^d)$ where $N$ is the number of objects and $d$ is the typical interaction clique size. Fundamental physics interactions simulated by the method have $d = 2$, resulting in a quadratic dependence and higher order interactions become completely unmanageable. In Social LSTM [10], this was remedied by pooling a local neighborhood of interactions. The solution however cannot work for scenarios with long-range interactions. Another solution offered by Battaglia et al. [9] is to add several fully connected layers modeling the high-order interactions. This approach struggles when the objective is to select one of the agents (e.g. which agent will move), as it results in a distributed representation and loses the structure of the problem.

In this work we present VAIN (Vertex Attention Interaction Network), a novel multi-agent attentional neural network for predictive modeling. VAIN's attention mechanism helps with modeling the locality of interactions and improves performance by determining which agents will share information. VAIN can be said to be a CommNet [11] with a novel attention mechanism or a factorized Interaction Network [9]. This will be made more concrete in Sec. 2. We show that VAIN can model high-order interactions with linear complexity in the number of vertices while preserving the structure of the problem, this has lower complexity than IN in cases where there are many fewer vertices than edges (in many cases linear vs quadratic in the number of agents).

For evaluation we introduce two non-physical tasks which more closely resemble real-world and game-playing multi-agent predictive modeling, as well as a physical Bouncing Balls task. Our non-physical tasks are taken from Chess and Soccer and contain different types of interactions and different data regimes. The interaction graph on these tasks is not known apriori, as is typical in nature.

An informal analysis of our architecture is presented in Sec. 2. Our method is presented in Sec. 3. Description of our experimental evaluation scenarios and our results are provided in Sec. 4. Conclusion and future work are presented in Sec. 5.

**Related Work**

This work is primarily concerned with learning multi-agent interactions with graph structures. The seminal works in graph neural networks were presented by Scarselli et al. [12, 13] and Li et al. [14]. Another notable iterative graph-like neural algorithm is the Neural-GPU [15]. Notable works in graph NNs includes Spectral Networks [16] and work by Duvenaud et al. [17] for fingerprinting of chemical molecules.

Two related approaches that learn multi-agent interactions on a graph structure are: Interaction Networks [9] which learn a physical simulation of objects that exhibit binary relations and Communication Networks (CommNets) [11], presented for learning optimal communications between agents. The differences between our approach VAIN and previous approaches INs and CommNets are analyzed in detail in Sec. 2.

Another recent approach is PointNet [18] where every point in a point cloud is embedded by a deep neural net, and all embeddings are pooled globally. The resulting descriptor is used for classification and segmentation. Although a related approach, the paper is focused on 3D point clouds rather than multi-agent systems. A different approach is presented by Social LSTM [10] which learns social interaction by jointly training multiple interacting LSTMs. The complexity of that approach is quadratic in the number of agents requiring the use of local pooling that only deals with short range interactions to limit the number of interacting bodies.

The attentional mechanism in VAIN has some connection to Memory Networks [19, 20] and Neural Turning Machines [21]. Other works dealing with multi-agent reinforcement learning include [22] and [23].

There has been much work on board game bots (although the approach of modeling board games as interactions in a neural network multi agent system is new). Approaches include [24, 25] for Chess, [26, 27, 28] for Backgammons [29] for Go.

*Concurrent work:* We found on Arxiv two concurrent submissions which are relevant to this work. Santoro et al. [30] discovered that an architecture nearly identical to Interaction Nets achieves excellent performance on the CLEVR dataset [31]. We leave a comparison on CLEVR for future work. Vaswani et al. [32] use an architecture that bears similarity to VAIN for achieving state-of-the-art performance for machine translation. The differences between our work and Vaswani et al.'s concurrent work are substantial in application and precise details.

## 2 Factorizing Multi-Agent Interactions

In this section we give an informal analysis of the multi-agent interaction architectures presented by Interaction Networks [9], CommNets [11] and VAIN.

Interaction Networks model each interaction by a neural network. For simplicity of analysis, let us restrict the interactions to be of 2nd order. Let $\psi_{int}(x_i, x_j)$ be the interaction between agents $A_i$ and $A_j$, and $\phi(x_i)$ be the non-interacting features of agent $A_i$. The output is given by a function $\theta()$ of the sum of all of the interactions of $A_i$, $\sum_j \psi_{int}(x_i, x_j)$ and of the non-interacting features $\phi(x_i)$.

$$o_i = \theta(\sum_{j \neq i} \psi_{int}(x_i, x_j), \phi(x_i)) \tag{1}$$

A single step evaluation of the output for the entire system requires $O(N^2)$ evaluations of $\psi_{int}()$.

An alternative architecture is presented by CommNets, where interactions are not modeled explicitly. Instead an interaction vector is computed for each agent $\psi_{com}(x_i)$. The output is computed by:

$$o_i = \theta(\sum_{j \neq i} \psi_{com}(x_j), \phi(x_i)) \tag{2}$$

A single step evaluation of the CommNet architecture requires $O(N)$ evaluations of $\psi_{com}()$. A significant drawback of this representation is not explicitly modeling the interactions and putting the whole burden of modeling on $\theta$. This can often result in weaker performance (as shown in our experiments).

VAIN's architecture preserves the complexity advantages of CommNet while addressing its limitations in comparison to IN. Instead of requiring a full network evaluation for every interaction pair $\psi_{int}(x_i, x_j)$ it learns a communication vector $\psi_{vain}^c(x_i)$ for each agent and additionally an attention vector $a_i = \psi_{vain}^a(x_i)$. The strength of interaction between agents is modulated by kernel function $e^{|a_i - a_j|^2}$. The interaction is approximated by:

$$\psi_{int}(x_i, x_j) = e^{|a_i - a_j|^2} \psi_{vain}(x_j) \tag{3}$$

The output is given by:

$$o_i = \theta(\sum_{j \neq i} e^{|a_i - a_j|^2} \psi_{vain}(x_j), \phi(x_i)) \tag{4}$$

In cases where the kernel function is a good approximation for the relative strength of interaction (in some high-dimensional linear space), VAIN presents an efficient linear approximation for IN which preserves CommNet's complexity in $\psi()$.

Although physical interactions are often additive, many other interesting cases (Games, Social, Team Play) are not additive. In such cases the average instead the sum of $\psi$ should be used (in [9] only physical scenarios were presented and therefore the sum was always used, whereas in [11] only non-physical cases were considered and therefore only averaging was used). In non-additive cases VAIN uses a softmax:

$$K_{i,j} = e^{|a_i - a_j|^2} / \sum_j e^{|a_i - a_j|^2} \tag{5}$$

## 3 Model Architecture

In this section we model the interaction between $N$ agents denoted by $A_1...A_N$. The output can be either be a prediction for every agent or a system-level prediction (e.g. predict which agent will act

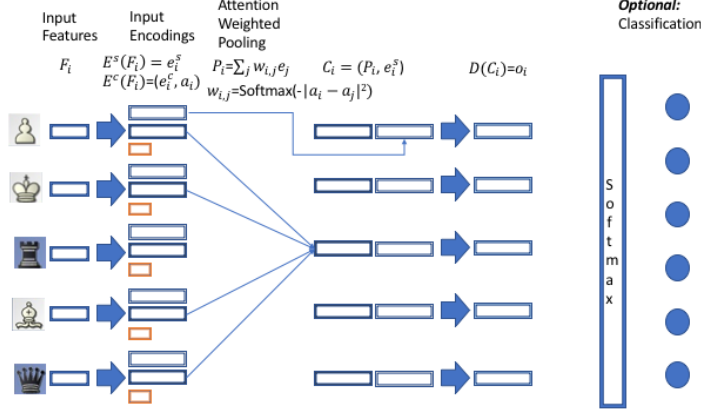

Figure 1: A schematic of a single-hop VAIN: i) The agent features $F_i$ are embedded by singleton encoder $E^s()$ to yield encoding $e_i^s$ and communications encoder $E^c()$ to yield vector $e_i^c$ and attention vector $a_i$ ii) For each agent an attention-weighted sum of all embeddings $e_i^c$ is computed $P_i = \sum_j w_{i,j} * e_j^c$. The attention weights $w_{i,j}$ are computed by a Softmax over $-||a_i - a_j||^2$. The diagonal $w_{i,i}$ is set to zero to exclude self-interactions. iii) The singleton codes $e_i^s$ are concatenated with the pooled feature $P_i$ to yield intermediate feature $C_i$ iv) The feature is passed through decoding network $D()$ to yield per-agent vector $o_i$. For Regression: $o_i$ is the final output of the network. vii) For Classification: $o_i$ is scalar and is passed through a Softmax.

next). Although it is possible to use multiple hops, our presentation here only uses a single hop (and they did not help in our experiments).

Features are extracted for every agent $A_i$ and we denote the features by $F_i$. The features are guided by basic domain knowledge (such as agent type or position).

We use two agent encoding functions: i) a singleton encoder for single-agent features $E^s()$ ii) A communication encoder for interaction with other agents $E^c()$. The singleton encoding function $E^s()$ is applied on all agent features $F_i$ to yield singleton encoding $e_i^s$

$$E^s(F_i) = e_i^s \tag{6}$$

We define the communication encoding function $E^c()$. The encoding function is applied to all agent features $F_i$ to yield both encoding $e_i^c$ and attention vector $a_i$. The attention vector is used for addressing the agents with whom information exchange is sought. $E^c()$ is implemented by fully connected neural networks (from now FCNs).

$$E^c(F_i) = (e_i^c, a_i) \tag{7}$$

For each agent we compute the pooled feature $P_i$, the interaction vectors from other agents weighted by attention. We exclude self-interactions by setting the self-interaction weight to 0:

$$P_i = \sum_j e_j * Softmax(-||a_i - a_j||^2) * (1 - \delta_{j=i}) \tag{8}$$

This is in contrast to the average pooling mechanism used in CommNets and we show that it yields better results. The motivation is to average only information from relevant agents (e.g. nearby or particularly influential agents). The weights $w_{i,j} = Softmax_j(-||a_i - a_j||^2)$ give a measure of the interaction between agents. Although naively this operation scales quadratically in the number of agents, it is multiplied by the feature dimension rather by a full $E()$ evaluation and is therefore significantly smaller than the cost of the (linear number) of $E()$ calculations carried out by the algorithm. In case the number of agents is very large (>1000) the cost can still be mitigated: The Softmax operation often yields a sparse matrix, in such cases the interaction can be modeled by the K-Nearest neighbors (measured by attention). The calculation is far cheaper than evaluating $E^c()$

$O(N^2)$ times as in IN. In cases where even this cheap operation is too expensive we recommend to using CommNets as a default as they truly have an O(N) complexity.

The pooled-feature $P_i$ is concatenated to the original features $F_i$ to form intermediate features $C_i$:

$$C_i = (P_i, e_i) \tag{9}$$

The features $C_i$ are passed through decoding function $D()$ which is also implemented by FCNs. The result is denoted by $o_i$:

$$o_i = D(C_i) \tag{10}$$

For regression problems, $o_i$ is the per-agent output of VAIN. For classification problems, $D()$ is designed to give scalar outputs. The result is passed through a softmax layer yielding agent probabilities:

$$Prob(i) = Softmax(o_i) \tag{11}$$

Several advantages of VAIN over Interaction Networks [9] are apparent:

*Representational Power:* VAIN does not assume that the interaction graph is pre-specified (in fact the attention weights $w_{i,j}$ learn the graph). Pre-specifying the graph structure is advantageous when it is clearly known e.g. spring-systems where locality makes a significant difference. In many multi-agent scenarios the graph structure is not known apriori. Multiple-hops can give VAIN the potential to model higher-order interactions than IN, although this was not found to be advantageous in our experiments.

*Complexity:* As explained in Sec. 2, VAIN features better complexity than INs. The complexity advantage increases with the order of interaction.

## 4   Evaluation

We presented VAIN, an efficient attentional model for predictive modeling of multi-agent interactions. In this section we show that our model achieves better results than competing methods while having a lower computational complexity.

We perform experiments on tasks from different multi-agent domains to highlight the utility and generality of VAIN: chess move, soccer player prediction and physical simulation.

### 4.1   Chess Piece Prediction

Chess is a board game involving complex multi-agent interactions. Chess is difficult from a multi-agent perspective due to having 12 different types of agents and non-local high-order interactions. In this experiment we do not attempt to create an optimal chess player. Rather, we are given a board position from a professional game. Our task is to identify the piece that will move next (MPP). There are 32 possible pieces, each encoded by one-hot encodings of $piecetype$, $x$ position, $y$ position. Missing pieces are encoded with all zeros. The output is the $id$ of the piece that will move next.

For training and evaluation of this task we downloaded 10k games from the FICS Games Dataset, an on-line repository of chess games. All the games used are standard games between professionally ranked players. 9k randomly sampled games were used for training, and the remaining 1k games for evaluation. Moves later in the game than 100 (i.e. 50 Black and 50 White moves), were dropped from the dataset so as not to bias it towards particularly long games. The total number of moves is around 600k.

We use the following methods for evaluation: $Rand$: Random piece selection. $FC$: A standard FCN with three hidden layers (64 hidden nodes each). This method requires indexing to be learned. $SMax$: Each piece is encoded by neural network into a scalar "vote". The "votes" from all input pieces are fed to a $Softmax$ classifier predicting the output label. This approach does not require learning to index, but cannot model interactions. $1hop - FC$: Each piece is encoded as in SMax but to a vector rather than a scalar. A deep (3 layer) classifier predicts the MPP from the concatenation of the vectors. $CommNet$: A standard CommNet (no attention) [11]. The protocol for CommNet is the same as VAIN. $IN$: An Interaction Network followed by Softmax (as for VAIN). Inference for this IN required around 8 times more computation than VAIN and CommNet. $ours - VAIN$.

Table 1: Accuracy (%) for the Next Moving Piece (MPP) experiments.

| $Rand$ | $FC$ | $SMax$ | $1hop-FC$ | $CommNet$ | $IN$ | $ours$ |
|---|---|---|---|---|---|---|
| 4.5 | 21.6 | 13.3 | 18.6 | 27.2 | 28.3 | **30.1** |

The results for next moving chess piece prediction can be seen in Table. 1. Our method clearly outperforms the competing baselines illustrating that VAIN is effective at selection type problems - i.e. selecting 1 - of- $N$ agents according to some criterion (in this case likelihood to move). The non-interactive method $SMax$ performs much better than $Rand$ (+9%) due to use of statistics of moves. Interactive methods ($FC$, $1hot-FC$, $CommNet$, $IN$ and $VAIN$) naturally perform better as the interactions between pieces are important for deciding the next mover. It is interesting that the simple $FC$ method performs better than $1hop-FC$ (+3%), we think this is because the classifier in $1hop-FC$ finds it hard to recover the indexes after the average pooling layer. This shows that one-hop networks followed by fully connected classifiers (such as the original formulation of Interaction Networks) struggle at selection-type problems. Our method $VAIN$ performs much better than $1hop-IN$ (11.5%) due to the per-vertex outputs $o_i$, and coupling between agents. $VAIN$ also performs significantly better than $FC$ (+8.5%) as it does not have to learn indexing. It outperforms vanilla CommNet by 2.9%, showing the advantages of our attentional mechanism. It also outperforms INs followed by a per-agent Softmax (similarly to the formulation for VAIN) by 1.8% even though the IN performs around 8 times more computation than VAIN.

## 4.2 Soccer Players

Team-player interaction is a promising application area for end-to-end multi-agent modeling as the rules of sports interaction are quite complex and not easily formulated by hand-coded rules. An additional advantage is that predictive modeling can be self-supervised and no labeled data is necessary. In team-play situations many agents may be present and interacting at the same time making the complexity of the method critical for its application.

In order to evaluate the performance of VAIN on team-play interactions, we use the Soccer Video and Player Position Dataset (SVPP) [33]. The SVPP dataset contains the parameters of soccer players tracked during two home matches played by Tromsø IL, a Norwegian soccer team. The sensors were positioned on each home team player, and recorded the player's location, heading direction and movement velocity (as well as other parameters that we did not use in this work). The data was re-sampled by [33] to occur at regular 20 Hz intervals. We further subsampled the data to 2 Hz. We only use sensor data rather than raw-pixels. End-to-end inference from raw-pixel data is left to future work.

The task that we use for evaluation is predicting from the current state of all players, the position of each player for each time-step during the next 4 seconds (i.e. at $T+0.5$, $T+1.0$ ... $T+4.0$). Note that for this task, we just use a single frame rather than several previous frames, and therefore do not use RNN encoders for this task.

We use the following methods for evaluation: $Static$: trivial prediction of 0-motion. $PALV$: Linearly extrapolating the agent displacement by the current linear velocity. $PALAF$: A linear regressor predicting the agent's velocity using all features including the velocity, but also the agent's heading direction and most significantly the agent's current field position. $PAD$: a predictive model using all the above features but using three fully-connected layers (with 256, 256 and 16 nodes). $CommNet$: A standard CommNet (no attention) [11]. The protocol for CommNet is the same as VAIN. $IN$: An Interaction Network [9], requiring $O(N^2)$ network evaluations. $ours$: VAIN.

We excluded the second half of the Anzhi match due to large sensor errors for some of the players (occasional 60m position changes in 1-2 seconds).

A few visualizations of the Soccer scenario can be seen in Fig. 4. The positions of the players are indicated by green circles, apart from a target player (chosen by us), that is indicated by a blue circle. The brightness of each circle is chosen to be proportional to the strength of attention between each player and the target player. Arrows are proportional to player velocity. We can see in this scenario that the attention to nearest players (attackers to attackers, midfielder to midfielders) is strongest, but attention is given to all field players. The goal keeper normally receives no attention (due to being

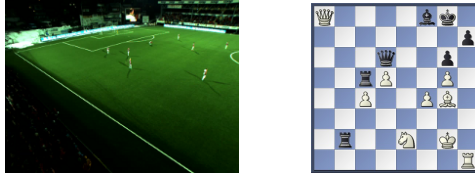

Figure 2: a) A soccer match used for the Soccer task. b) A chess position illustrating the high-order nature of the interactions in next move prediction. Note that in both cases, VAIN uses agent positional and sensor data rather than raw-pixels.

Table 2: Soccer Prediction errors (meters).

| Experiments | | Methods | | | | | | |
|---|---|---|---|---|---|---|---|---|
| Dataset | Time-step | $Static$ | $PALV$ | $PALAF$ | $PAD$ | $IN$ | $CommNet$ | $ours$ |
| | 0.5 | 0.54 | **0.14** | **0.14** | **0.14** | 0.16 | 0.15 | **0.14** |
| 1103a | 2.0 | 1.99 | 1.16 | 1.14 | 1.13 | **1.09** | 1.10 | **1.09** |
| | 4.0 | 3.58 | 2.67 | 2.62 | 2.58 | **2.47** | 2.48 | **2.47** |
| | 0.5 | 0.49 | **0.13** | **0.13** | **0.13** | 0.14 | **0.13** | **0.13** |
| 1103b | 2.0 | 1.81 | 1.06 | 1.06 | 1.04 | **1.02** | **1.02** | **1.02** |
| | 4.0 | 3.27 | 2.42 | 2.41 | 2.38 | **2.30** | 2.31 | **2.30** |
| | 0.5 | 0.61 | **0.17** | **0.17** | **0.17** | **0.17** | **0.17** | **0.17** |
| 1107a | 2.0 | 2.23 | 1.36 | 1.34 | 1.32 | 1.26 | 1.26 | **1.25** |
| | 4.0 | 3.95 | 3.10 | 3.03 | 2.99 | 2.82 | 2.81 | **2.79** |
| Mean | | 1.84 | 1.11 | 1.10 | 1.08 | 1.04 | 1.04 | **1.03** |

far away, and in normal situations not affecting play). This is an example of mean-field rather than sparse attention.

We evaluated our methods on the SVPP dataset. The prediction errors in Table. 2 are broken down for different time-steps and for different train / test datasets splits. It can be seen that the non-interactive baselines generally fare poorly on this task as the general configuration of agents is informative for the motion of agents beyond a simple extrapolation of motion. Examples of patterns than can be picked up include: running back to the goal to help the defenders, running up to the other team's goal area to join an attack. A linear model including all the features performs better than a velocity only model (as position is very informative). A non-linear per-player model with all features improves on the linear models. The interaction network, CommNet and VAIN significantly outperform the non-interactive methods. VAIN outperformed CommNet and IN, achieving this with only 4% of the number of encoder evaluations performed by IN. This validates our premise that VAIN's architecture can model object interactions without modeling each interaction explicitly.

## 4.3 Bouncing Balls

Following Battaglia et al. [9], we present a simple physics-based experiment. In this scenario, balls are bouncing inside a 2D square container of size $L$. There are $N$ identical balls (we use $N = 50$) which are of constant size and are perfectly elastic. The balls are initialized at random positions and with initial velocities sampled at random from $[-v_0..v_0]$ (we use $v_0 = 3ms^{-1}$). The balls collide with other balls and with the walls, where the collisions are governed by the laws of elastic collisions. The task which we evaluate is the prediction of the displacement and change in velocity of each ball in the next time step. We evaluate the prediction accuracy of our method $VAIN$ as well as Interaction Networks [9] and CommNets [11]. We found it useful to replace VAIN's attention mechanism by an unnormalized attention function due to the additive nature of physical forces:

$$p_{i,j} = e^{-||a_i - a_j||^2} - \delta_{i,j} \tag{12}$$

In Fig. 4 we can observe the attention maps for two different balls in the Bouncing Balls scenario. The position of the ball is represented by a circle. The velocity of each ball is indicated by a line

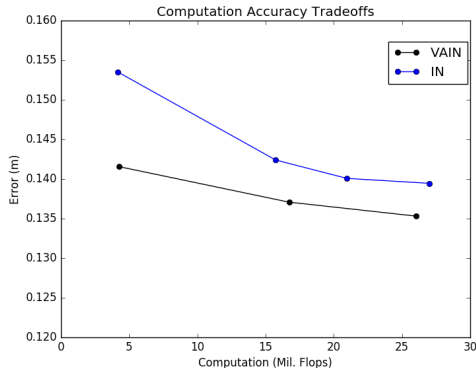

Figure 3: Accuracy differences between VAIN and IN for different computation budgets: VAIN outperforms IN by spending its computation budget on a few larger networks (one for each agent) rather than many small networks (one for every pair of agents). This is even more significant for small computation budgets.

Table 3: RMS accuracy of Bouncing Ball next step prediction.

|  | *VEL0* | *VEL-CONST* | *COMMNET* | *IN* | *VAIN* |
|---|---|---|---|---|---|
| $RMS$ | 0.561 | 0.547 | 0.510 | 0.139 | **0.135** |

extending from the center of the circle, the length of the line is proportional to the speed of the ball. For each figure we choose a target ball $A_i$, and paint it blue. The attention strength of each agent $A_j$ with respect to $A_i$ is indicated by the shade of the circle. The brighter the circle, the stronger the attention. In the first scenario we observe that the two balls near the target receive attention whereas other balls are suppressed. This shows that the system exploits the sparsity due to locality that is inherent to this multi-agent system. In the second scenario we observe, that the ball on a collision course with the target receives much stronger attention, relative to a ball that is much closer to the target but is not likely to collide with it. This indicates VAIN learns important attention features beyond the simple positional hand-crafted features typically used.

The results of our bouncing balls experiments can be seen in Tab. 3. We see that in this physical scenario VAIN significantly outperformed CommNets, and achieves better performance than Interaction Networks for similar computation budgets. In Fig. 4.2 we see that the difference increases for small computation budgets. The attention mechanism is shown to be critical to the success of the method.

## 4.4 Analysis and Limitations

Our experiments showed that VAIN achieves better performance than other architectures with similar complexity and equivalent performance to higher complexity architectures, mainly due to its attention mechanism. There are two ways in which the attention mechanism implicitly encodes the interactions of the system: i) Sparse: if only a few agents significantly interact with agent $a_o$, the attention mechanism will highlight these agents (finding $K$ spatial nearest neighbors is a special case of such attention). In this case CommNets will fail. ii) Mean-field: if a space can be found where the important interactions act in an additive way, (e.g. in soccer team dynamics scenario), the attention mechanism would find the correct weights for the mean field. In this case CommNets would work, but VAIN can still improve on them.

VAIN is less well-suited for cases where both: interactions are not sparse such that the K most important interactions will not give a good representation and where the interactions are strong and highly non-linear so that a mean-field approximation is non-trivial. One such scenario is the $M$ body gravitation problem. Interaction Networks are particularly well suited for this scenario and VAIN's factorization will not yield an advantage.

**Implementation**

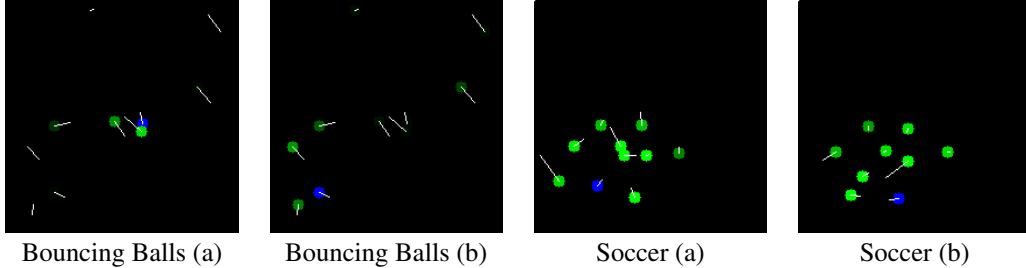

| Bouncing Balls (a) | Bouncing Balls (b) | Soccer (a) | Soccer (b) |

Figure 4: A visualization of attention in the Bouncing Balls and Soccer scenarios. The target ball is blue, and others are green. The brightness of each ball indicates the strength of attention with respect to the (blue) target ball. The arrows indicate direction of motion. *Bouncing Balls*: Left image: The ball nearer to target ball receives stronger attention. Right image: The ball on collision course with the target ball receives much stronger attention than the nearest neighbor of the target ball. *Soccer*: This is an example of mean-field type attention, where the nearest-neighbors receive privileged attention, but also all other field players receive roughly equal attention. The goal keeper typically receives no attention due to being far away.

*Soccer*: The encoding and decoding functions $E_c()$, $E_s()$ and $D()$ were implemented by fully-connected neural networks with two layers, each of 256 hidden units and with ReLU activations. The encoder outputs had 128 units. For IN each layer was followed by a BatchNorm layer (otherwise the system converged slowly to a worse minimum). For VAIN no BatchNorm layers were used. *Chess*: The encoding and decoding functions $E()$ and $D()$ were implemented by fully-connected neural networks with three layers, each of width 64 and with ReLU activations. They were followed by BatchNorm layers for both IN and VAIN. *Bouncing Balls*: The encoding and decoding function $E_c()$, $E_s()$ and $D()$ were implemented with FCNs with 256 hidden units and three layer. The encoder outputs had 128 units. No BatchNorm units were used. For Soccer, $E_c()$ and $D()$ architectures for VAIN and IN was the same. For Chess we evaluate INs with $E_c()$ being 4 times smaller than for VAIN, this still takes 8 times as much computation as used by VAIN. For Bouncing Balls the computation budget was balanced between VAIN and IN by decreasing the number of hidden units in $E_c()$ for IN by a constant factor.

In all scenarios the attention vector $a_i$ is of dimension 10 and shared features with the encoding vectors $e_i$. Regression problems were trained with $L2$ loss, and classification problems were trained with cross-entropy loss. All methods were implemented in PyTorch [34] in a Linux environment. End-to-end optimization was carried out using ADAM [35] with $\alpha = 1e-3$ and no $L2$ regularization was used. The learning rate was halved every 10 epochs. The chess prediction training for the MPP took several hours on a M40 GPU, other tasks had shorter training times due to smaller datasets.

## 5 Conclusion and Future Work

We have shown that VAIN, a novel architecture for factorizing interaction graphs, is effective for predictive modeling of multi-agent systems with a linear number of neural network encoder evaluations. We analyzed how our architecture relates to Interaction Networks and CommNets. Examples were shown where our approach learned some of the rules of the multi-agent system. An interesting future direction to pursue is interpreting the rules of the game in symbolic form, from VAIN's attention maps $w_{i,j}$. Initial experiments that we performed have shown that some chess rules can be learned (movement of pieces, relative values of pieces), but further research is required.

## Acknowledgement

We thank Rob Fergus for significant contributions to this work. We also thank Gabriel Synnaeve and Arthur Szlam for fruitful comments on the manuscript.

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
