[Reviews · NeurIPS 2017]

Reviewer 1



This paper extends interaction networks (INs) with an attentional mechanism so that it scales linearly (as opposed to quadratically in vanilla INs) with the number of agents in a multi-agent predictive modeling setting: the embedding network is evaluated once per agent rather than once for every interaction. This allows to model higher-order interactions between agents in a computationally efficient way. The method is evaluated on two new non-physical tasks of predicting chess piece selection and soccer player movements. The paper proposes a simple and elegant attentional extension of Interaction Networks, and convincingly shows the benefit of the approach with two interesting experiments. The idea is not groundbreaking but seems sufficiently novel, especially in light of its effectiveness. The paper is clear and well written. I appreciate that the authors provide an intuitive yet precise explanation (in words) of their mechanism instead of resorting to unnecessarily complicated notation and formulas. Some confusion comes from the discussions about single-hop and multi-hop, terms that should probably be defined somewhere. On line 105, the authors write: "The architecture can be described as a single-hop CommNet with attention. Although CommNets typically use more hops for learning communications, more than one hop did not help on our predictive modeling tasks", but on lines 136 and 262 it says that VAIN uses multiple hops. The experiments are well thought-out and interesting. I'm only missing some more insights or visualizations on what kind of interactions the attention mechanism learns to focus on or ignore, as the attention mechanism is the main contribution of the paper. Further, I wonder if a disadvantage of the proposed attention mechanism (with the fixed-length softmax layer depending on the number of agents) does not take away one of the benefits of vanilla INs, namely that it is straightforward to dynamically add or remove agents during inference. If so, perhaps this can be mentioned. Some minor remarks and typo's: - typo in sentence on line 103-104: two times "be" - equation (3) should be C = (P_i, F_i), or the preceding sentence should be changed - line 138: sentence should start with "*While* the original formulation ..." - line 284: generally *fares* poorly. Overall, I believe this work would be a valuable contribution to NIPS.

Reviewer 2



It's not clear to me what the utility of the claimed predictive modeling is. The prediction task in the chess domain of predicting the piece that will move next (MPP) and the next piece that will be taken (TPP) seems strange to me. I don't understand the connection with reinforcement learning that the authors were describing in line 90. The caveat here is that I'm not familiar with CommNets, which the current work is supposed to based on. However, independent from that, I think the paper will benefit from a clearer exposition of the particular tasks and evaluation metrics. Similar to the chess domain, I also don't fully understand what the model is supposed to predict for the soccer domain. If the task is to predict the positions of all players for the next 4 seconds based on the current frame, it's not clear to me how meaningful the task is. What role to other contexts, such as the positions of the opposing team, for example, play in this prediction model?

Reviewer 3



Summary: The authors propose to model predictive tasks in board games via multi-agent interactions. The presented architecture is a CommNet where the interactions P_i affecting actor i are computed by a weighted average instead of a uniform average. The weights w_{i,j} are produced by a softmax over all pair-wise products between the attention vector of actor i with another actor. After learning, the weights encode the graph structure, i.e. which agents affect each other. The interactions P_i are then concatenated with the input features and processed by an MLP for regression (e.g. prediction of a soccer player’s field position in the near future) or classification (e.g. which chess piece moves next or is taken in the next move). The authors’ premise that the model can learn interaction patterns although evaluating the embedding network per agent (instead of per interaction), is validated in the experimental section. Qualitative Assessment: The paper reads well and is easy to follow. The experimental setup is clear and provides almost enough details for replication. The following points are missing, as far as I can tell: - I can’t seem to find the number of steps or epochs, the model’s were trained for. There is also no mention of early stopping. How did you decide to stop the training? Was there a threshold for convergence of the training accuracy? - How did you decide not to use BatchNorm in the soccer experiments, but to add it in the chess experiments? - Are the numbers reported in Table 1 and 2 single runs or averaged over multiple runs? It is often advisable to display the mean and standard deviation or the min and max over multiple runs, especially since the differences between some rows in the soccer table are very small. Another point I would like to see clarified is the claim in line 99 that modeling board games as interactions between agents is novel. Isn’t that what most of game theory is about? In line 272, I would change the sentence to "It is clear that VAIN and CommNet perform…" (as CommNet without pretraining performs better than VAIN in the middle column of table 1)